# Learning without Prejudices: Continual Unbiased Learning via Benign and Malignant Forgetting

**Myeongho Jeon**[*], **Hyoje Lee**[*], **Yedarm Seong, Myungjoo Kang**
Seoul National University
`{andyjeon, hyoje42, mybirth0407, mkang}@snu.ac.kr`

## Abstract

Although machine learning algorithms have achieved state-of-the-art status in image classification, recent studies have substantiated that the ability of the models to learn several tasks in sequence, termed continual learning (CL), often suffers from abrupt degradation of performance from previous tasks. A large body of CL frameworks has been devoted to alleviating this forgetting issue. However, we observe that forgetting phenomena in CL are not always unfavorable, especially when there is bias (spurious correlation) in training data. We term such type of forgetting *benign forgetting*, and categorize detrimental forgetting as *malignant forgetting*. Based on this finding, our objective in this study is twofold: (a) to discourage malignant forgetting by generating previous representations, and (b) encourage benign forgetting by employing contrastive learning in conjunction with feature-level augmentation. Extensive evaluations of biased experimental setups demonstrate that our proposed method, *Learning without Prejudices*, is effective for continual unbiased learning.

## 1 Introduction

In continual learning (CL), a model learns a sequence of tasks to accumulate existing knowledge for a new task. This is preferable in practice, where a model cannot retrieve previously used data, owing to privacy, limited data capacity, or an online streaming setup. The main challenge in CL is to alleviate "catastrophic forgetting," whereby a model forgets prior information while training on new information (McCloskey & Cohen, 1989). A line of recent works has been dedicated to mitigating this issue. Regularization-based methods force a current model not to be far from the previous one by penalizing changes in the parameters learned in previous tasks (Kirkpatrick et al., 2017; Chaudhry et al., 2018; Aljundi et al., 2018; 2019a; Ahn et al., 2019; Dhar et al., 2019; Douillard et al., 2020). Replay-based methods store samples of prior tasks in a buffer and employ them along with present samples (Robins, 1995; Lopez-Paz & Ranzato, 2017; Buzzega et al., 2020; Aljundi et al., 2019b; Mai et al., 2021; Lin et al., 2021; Madaan et al., 2021; Chaudhry et al., 2021; Bonicelli et al., 2022). Generator-based methods generate prior samples and input them into current tasks (Shin et al., 2017; Kemker & Kanan, 2017; Xiang et al., 2019; Ostapenko et al., 2019; Liu et al., 2020; Yin et al., 2020).

A common assumption of the above-mentioned existing methods is that the training dataset is well-distributed. However, a source dataset is often biased, and a machine learning algorithm could perceive the bias as meaningful information, thereby leading to misleading generalizability of the model (Kim et al., 2019; Jeon et al., 2022). In the experiment in Section 3.1, we show that biased distributions are detrimental to the robustness of models in existing CL scenarios. Thus, we propose a new type of CL, termed "continual unbiased learning (CUL)", in which the dataset of each task has a different bias. With CUL, we aim to make any model trained on any task unbiased, considering all models as candidates for application. This is particularly desirable in practice, whereby a model designed for a specific purpose is deployed for long periods and training datasets with divergent distributions are fed sequentially to update the model.

---

[*]Equal contribution

Even with CUL, forgetting past information ("malignant forgetting") degrades the generalizability of a model. For instance, with Biased MNIST in Figure 1, the classifier perceives color as meaningful information for prediction, although it is not a natural meaning associated with the number. If the model clearly memorizes prior information that there are (red, 0) and (gray, 0) samples, it could know that color is not the key factor for predicting numbers. Furthermore, we observe that *forgetting is not always malignant* through the experiment in Section 3.2. Although information (derived from prior data) itself can contribute to a model's generalizability, it is beneficial to forget the misguidance learned from biased datasets, and hence we term such a forgetting "benign forgetting". As an example, suppose a classifier trained on the MNIST dataset is extremely biased toward the background color, as in Section 3.2. It is unfavorable for the classifier to make a logic that $color = number$ and thus bet all the 'blue' images on '3', for instance.

Therefore, we aim to discourage *malignant forgetting* and encourage *benign forgetting*. Toward this, we design a novel method, named `Learning without Prejudices (LwP)`, which employs *feature generator* and *contrastive learning*. (*i*) Inspired by the research in Section 3.1 that the model trained with a set of data from all the tasks does not suffer from *malignant forgetting*, we exploit the capabilities of a *feature generator*. The *feature generator* generates feature maps containing previous information via a generative adversarial network (GAN). Feature maps provide a larger range of feature space (to be referenced to) than images, making the classifier more robust. (*ii*) The generated features are fed into the model by *contrastive learning* (Grill et al., 2020), and then current data are used for training in supervised mode. Because bias means a spurious correlation between some particular attribute variables and label space, the model can learn representations free of bias, with self-supervised learning that does not require labels. (*iii*) To optimize the classifier with generated features effectively, we propose *feature-level augmentation* that spatially and channel-wise transforms features. An extensive evaluation of biased datasets shows that our proposed framework is effective for CUL. The main contributions of this study are summarized as follows:

- We present a novel framework, termed "continual unbiased learning", to address bias in CL. Additionally, we propose continual unbiased learning benchmarks and an evaluation protocol for future research.

- We find that forgetting phenomena in CL is not always catastrophic when the training dataset exhibits the non-uniform distribution of features, *e.g.*, a biased dataset, and hence categorize them into *malignant forgetting* and *benign forgetting*.

- We propose a novel method, `Learning without Prejudices (LwP)`, that employs *a feature generator* and *contrastive learning*, presenting *feature-level augmentation* to bridge them. `LwP` contributes to models' generalizability significantly.

## 2 PRELIMINARIES

### 2.1 PROBLEM STATEMENT

**Bias.** Let $\mathcal{X}$ be an input space and $\mathcal{Y}$ be a label space. We define an attribute variable $attr$ as an informative data feature of $\mathbf{x} \in \mathcal{X}$, possibly ranging from fine details (*e.g.*, the pixel at $(0, 0)$ is black) to high-level semantics of the image (*e.g.*, there is a cat). Thus, a set of attributes can represent data $\mathbf{x}$. Formally, let $\mathcal{A}$ be an attribute space and $\alpha : \mathcal{X} \to 2^{\mathcal{A}}$, where $2^{\mathcal{A}}$ denotes the power set of $\mathcal{A}$. A function $\alpha$ extracts attribute variables $attr \in \mathcal{A}$ from input space $\mathcal{X}$, *i.e.*, $\alpha(\mathbf{x}) = \{attr_1, attr_2, \ldots, attr_n\}$. Among these $attr$, some might be very correlated to $\mathcal{Y}$ while they are irrelevant to the natural meaning of the target object. We define this $attr$ as "bias". As machine learning algorithms (*e.g.*, convolutional neural networks (CNNs)) are overly dependent on training data distribution, the model could be biased, potentially leading to misleading generalizability (Torralba & Efros, 2011; Tommasi et al., 2017; Jeon et al., 2022). For instance, according to Bahng et al. (2020), the majority of frog images are captured in swamp scenes and many bird images are captured in the sky, making the model consider the background as a dominating cue that often fails to infer (frog, sky) and (bird, swamp) images correctly.

**Continual learning.** Consider a dataset $\mathcal{D} = \{(\mathbf{x}, y) | \mathbf{x} \in \mathcal{X}, y \in \mathcal{Y}\}$ for a classification problem. "continual learning" is a learning type with a sequence of $\mathcal{D}_S = \{\mathcal{D}_t = (\mathcal{X}_t, \mathcal{Y}_t)\}_{t=1}^T$ where each $\mathcal{X}_t$ and $\mathcal{Y}_t$ implicitly changes, expecting that $f : \mathcal{X}_t \to \mathcal{Y}_t$ accumulates previous information without forgetting while learning new tasks. Here, $T$ means the number of tasks. A task $t$ is predicting the target label $y$ with unseen feature variable $\mathbf{x}$ and learning a task means the procedure of optimizing a classifier $f : \mathcal{X}_t \to \mathcal{Y}_t$ with $\mathcal{D}_t$ to make a discriminative logic.

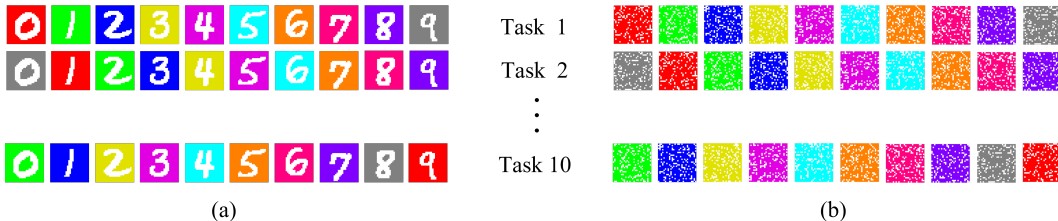

Task 1

Task 2

$\vdots$

Task 10

(a)                                                          (b)

Figure 1: Our experimental setup. (a) Biased MNIST. At each task, we change the bias (background color) sliding each one by +1 for the label (0-9), and the color corresponding to 9 moves to 0. (b) Distorted test set $\tilde{\mathcal{D}}_S$. We randomly choose pixels of $\alpha$ ratio on a single-colored image and then paint them to 'white' setting $\alpha$ as the average ratios of all the number pixels, which are represented by 'white', in the MNIST. A more detailed dataset configuration is provided in the Appendix.

**Continual unbiased learning.**   In addition to CL, we suppose $\mathcal{Y}_1 = \cdots = \mathcal{Y}_T$ and each $\mathcal{D}_t$ has a different bias. We aim at making the classifier $f$ unbiased toward any of $\mathcal{D}_t$ in the sequence, and term such type of learning as "continual unbiased learning". In practice, inconsistent distributions with different biases could be fed over time for a model that is applied for long periods for the same purpose. In this scenario, it is desirable for the model to be unbiased toward any of the datasets encompassing all of these domains.

## 2.2   Evaluating Protocol and Metrics

We set a bias attribute and define a dataset that has biases as a "biased dataset" and a dataset in which the bias attribute is uniformly distributed across the labels as an "unbiased dataset". We randomly split each biased dataset $\mathcal{D}_t = (\mathcal{D}_t^{train}, \mathcal{D}_t^{val}) \in \mathcal{D}_S$ into train and validation set to have the same ratio of biased samples. We use one unbiased test data $\mathcal{D}^{test}$ for evaluation. And, we define $f_t$ as the model trained with $\{\mathcal{D}_t^{train}\}_{t=1}^T$ sequentially and denote $\mathcal{F} = \{f_1, \cdots, f_T\}$.

After training the model with a sequence of the differently biased datasets in favor of one attribute or another, we evaluate its average accuracy following the conventional CL protocol: $\overline{Acc}(f_T, \mathcal{D}_S^{val}) = \frac{1}{T}\sum_{t=1}^T acc(f_T, \mathcal{D}_t^{val})$, where $T$ denotes the number of tasks and $acc(f_T, \mathcal{D}_t^{val})$ denotes the accuracy of the $T$-th model on the $t$-th task of the biased evaluation set. Additionally, we suggest an average unbiased accuracy for an unbiased test set whenever each task is trained to estimate the generalizability of all the models: $\overline{Acc}_{ub}(\mathcal{F}, \mathcal{D}^{test}) = \frac{1}{T}\sum_{t=1}^T acc(f_t, \mathcal{D}^{test})$, where $acc(f_t, \mathcal{D}^{test})$ denotes the accuracy of $f_t$ with unbiased test dataset. Using this metric, we can evaluate $\mathcal{F} = \{f_1, \cdots, f_T\}$, considering every model as a candidate for deployment.

## 3   Motivating Experiments

### 3.1   Continual learning on biased datasets

As a first step toward CUL, we investigate the learning tendency of a CNN-based classifier on biased datasets. We exploit biased MNIST (Bahng et al., 2020) toward background color, as shown in Figure 1. The unbiased test set has a uniform distribution for background color overall targets. Using the biased MNIST, we compare the average unbiased accuracy of the models trained with biased datasets $\cup_{1 \le t \le T}(\mathcal{D}_t)$ simultaneously, and in the sequence $\{\mathcal{D}_1, \cdots, \mathcal{D}_T\}$. Figure 2 (a) shows a significant performance gap between these two learning scenarios. We suggest that this is because the prior information is forgotten, *e.g.*, although the model refers to (red, 0) in the first task, it still makes a biased decision that $gray = 0$, considering only the (gray, 0) samples in the second task. We term this type of forgetting "malignant forgetting".

### 3.2   A closer look at forgetting

**Setup.**   With our motivation for the study being that a classifier could learn unintended information in biased conditions, hence forgetting such logic being desirable, we further conduct another motivating experiment. We additionally construct shape-absent color images and their label $(\tilde{\mathbf{x}}, y) \in \tilde{\mathcal{D}}$ to investigate the model's adaptability to a sequence of biased sets. The distorted dataset $\tilde{\mathcal{D}}$ is shown in Figure 1 (b). We estimate the *benign forgetting rate* (BFR), which quantifies the model's generalizability by calculating the forgetting rate of the previous biased logic during the training of

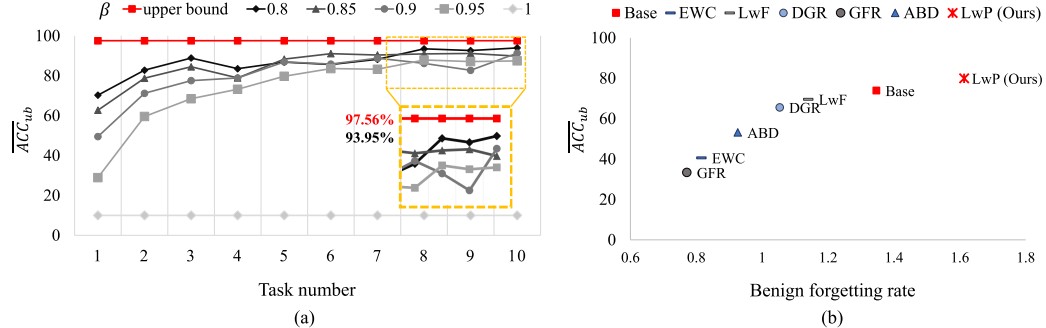

Figure 2: Motivating experiments. (a) Investigation for CUL. Each line in the graph means the performance of vanilla CNN for a sequence of Biased MNIST tasks with several different biased degrees $\beta$. Biased degree $\beta$ denotes the percentage of biased images in the dataset meaning '0.1' as an unbiased set with uniform distribution and '1' as a completely biased one. We train all the tasks from 1 to 10 concurrently and then represent its $\overline{Acc_{ub}}$ by red line indicating the upper bound. (b) Benign forgetting rate of models. We estimate BFR for the conventional CL model, the baseline, and our model (LwP).

new biased tasks. With all the models $\mathcal{F}$ and all the datasets $\tilde{\mathcal{D}}_S = \{\tilde{\mathcal{D}}_1, \cdots, \tilde{\mathcal{D}}_{T-1}\}$, BFR $\mathcal{B}(\mathcal{F}, \tilde{\mathcal{D}}_S)$ can be defined as

$$\mathcal{B}(\mathcal{F}, \tilde{\mathcal{D}}_S) = \frac{1}{T-1} \sum_{t=1}^{T-1} \frac{acc(f_t, \tilde{\mathcal{D}}_t) - acc(f_{t+1}, \tilde{\mathcal{D}}_t)}{acc(f_t, \tilde{\mathcal{D}}_t) - \frac{1}{n(\mathcal{Y}_t)}}, \tag{1}$$

where $acc(f_t, \tilde{\mathcal{D}}_t)$ denotes the accuracy of $f_t \in \mathcal{F}$ for $\tilde{\mathcal{D}}_t$, and $\tilde{\mathcal{D}}_t$ is the augmented images having the same correlation between the background and target labels as the original colored MNIST samples $\mathcal{D}_t$. The function $n(\mathcal{Y}_t)$ denotes the number of target labels, displaying 10 for MNIST (0-9). Intuitively, the performance distance $acc(f_t, \tilde{\mathcal{D}}_t) - 1/n(\mathcal{Y}_t)$ is the degree of the model's dependence on the bias attribute (color) for predicting $\mathcal{Y}_i$ because there is only 'color' information in $\tilde{\mathcal{D}}_t$ (Note that $1/n(\mathcal{Y}_t)$ is the lower limit of the model's discrimination, meaning $f_t$ takes a guess with unclear confidence). Thus, for the numerator $acc(f_t, \tilde{\mathcal{D}}_t) - acc(f_{t+1}, \tilde{\mathcal{D}}_t)$ with offset $1/n(\mathcal{Y}_t)$, if the classifier $f_{t+1}$ still sustains the biased logic of $f_t$ (e.g., $red = 0, \ldots, grey = 9$ for $f_1$ in Figure 1), the difference $acc(f_t, \tilde{\mathcal{D}}_t) - acc(f_{t+1}, \tilde{\mathcal{D}}_t) \approx 0$, whereas $acc(f_t, \tilde{\mathcal{D}}_t) - acc(f_{t+1}, \tilde{\mathcal{D}}_t)$ increases towards 1 otherwise. Denominator $acc(f_t, \tilde{\mathcal{D}}_t) - 1/n(\mathcal{Y}_t)$ is the scaling factor. This is because if the model is relatively unbiased and thus $acc(f_t, \tilde{\mathcal{D}}_t)$ is initially small, the numerator takes a penalty, making $acc(f_t, \tilde{\mathcal{D}}_t) - acc(f_{t+1}, \tilde{\mathcal{D}}_t)$ small (and $\mathcal{B}(\mathcal{F}, \tilde{\mathcal{D}}_S)$ becomes small), even though it forgets the biased logic well.

We set the baseline using a simple CNN model and compare it with conventional CL models, regularization-based (Kirkpatrick et al., 2017; Li & Hoiem, 2017), generator-based (Shin et al., 2017; Liu et al., 2020; Smith et al., 2021) methods. We calculate $\overline{Acc_{ub}}$ and $\mathcal{B}(\mathcal{F}, \tilde{\mathcal{D}}_S)$ with $\beta = 0.95$.

**Forgetting is not always malignant.** First, by performing the experiment displayed in Figure 2 (b), we find that the BFR and generalization on biased datasets have a correlation; meaning that *forgetting is not always catastrophic*. Confirming our assumption that forgetting previous biased logic is preferable in CUL, we define such forgetting as "benign forgetting". Second, it is notable that conventional CL methods are limited to CUL with insufficient BFR on biased MNIST, and unsatisfactory accuracy on the unbiased test set, even when compared to the baseline. This is because they are designed only to mitigate *malignant forgetting* and hence cannot adequately utilize *benign forgetting*.

## 4 METHODOLOGY

We discourage *malignant forgetting* by generating previous representations and encourage *benign forgetting* via contrastive learning. Although feature generator (Liu et al., 2020) and contrastive learning (Cha et al., 2021; Fini et al., 2022) are dedicated to CL, we first employ both of them, proposing feature-level augmentations to bridge the two methods. Additionally, we qualitatively and quantitatively suggest that these methods are effective for CUL.

### 4.1 Discouraging Malignant Forgetting

Let $f : \mathcal{X} \rightarrow \mathcal{Y}$ be a classifier with $L$ layers. For $1 \leq l < L$, classifier $f$ can be split into three sub-modules as $f = \{f_{[1,\dots,l]}, f_{[l+1,\dots,L-1]}, f_{[L]}\}$, where $[\cdot]$ denotes the indices of the layers included in the sub-modules. For convenience, we denote $f_{[1,\cdots,l]}$ as $f^a$ and $f_{[l+1,\cdots,L-1]}$ as $f^b$, *i.e.*, $f = \{f^a, f^b, f_{[L]}\}$.

To address *malignant forgetting*, we intend to make the latent feature vector of $l$-th layer $\mathbf{v} \in \mathbb{R}^{H \times W \times C}$ that include prior information. Following the adversarial training of the GAN (Gulrajani et al., 2017), we train *(feature) generator* $\mathcal{G} : \mathcal{Z} \rightarrow \mathbb{R}^{H \times W \times C}$ that maps the noise vector $\mathbf{z} \sim P_{\mathbf{z}}$ into $\mathbb{R}^{H \times W \times C}$, and discriminator $\mathcal{D} : \mathbb{R}^{H \times W \times C} \rightarrow [0, 1]$ that distinguishes real samples from $P_r$ and fake samples $\mathcal{G}(\mathbf{z})$. Thus, with fake features $\mathbf{v}_f := \mathcal{G}(\mathbf{z})$ and real features $\mathbf{v}_r \in \mathbb{R}^{H \times W \times C} := f^a(x)$, feature generator $\mathcal{G}$ and discriminator $\mathcal{D}$ are optimized by

$$\min_{\mathcal{G}} \max_{\mathcal{D}} \mathbb{E}_{\mathbf{x} \sim P_r}[\mathcal{D}(f^a(\mathbf{x}))] - \mathbb{E}_{\mathbf{z} \sim P_{\mathbf{z}}}[\mathcal{D}(\mathcal{G}(\mathbf{z}))]. \tag{2}$$

With $\mathcal{G}$ trained with prior tasks, $f^b$ receives both $\mathbf{v}_r$ (current) and $\mathbf{v}_f$ (previous) as input and hence can memorize previous knowledge, *i.e.*, discouraging *malignant forgetting*. When generating fake images and inputting them as Shin et al. (2017); Kemker & Kanan (2017); Xiang et al. (2019); Ostapenko et al. (2019), because they are only 3-channel aggregated features, the classifier may be confused if the generator does not work well. However, the generated feature maps have a larger range of feature space, providing the classifier with more information to be utilized selectively (please consult the comparison experiment in the Appendix.). This is favorable for CUL because a biased classifier results when the model is overly dependent on a certain few features, especially the bias attribute variable.

### 4.2 Encouraging Benign Forgetting

To train $f^b$, we apply bootstrap your own latent (BYOL) contrastive learning (Grill et al., 2020) because the generator $\mathcal{G}$ does not provide target labels for fake features. Although pseudo-labels can be obtained through a discriminator (Shin et al., 2017), auto-encoder-based generators (Kemker & Kanan, 2017), or conditional GANs (Xiang et al., 2019; Ostapenko et al., 2019), under biased conditions, few mislabeled samples could cause a large degradation in performance. Furthermore, bias is fundamentally based on the correlation between label and attribute space. Thus, label-free *contrastive learning* encourages the classifier to make representations independent of bias and hence to be robust for bias by forgetting the biased logic, *i.e.*, encouraging *benign forgetting*.

Following the two network designs of BYOL, let $g_{\theta}^{\text{online}} := \{f_{\theta}^b, p_{\theta}, q_{\theta}\}$ and $g_{\xi}^{\text{target}} := \{f_{\xi}^b, p_{\xi}\}$ be the online and target networks, where $p, q$ denote additional embedding layers and $\theta, \xi$ are the parameters of the online and target networks, respectively. The embedding layers $p$ encode $v$ and $q$ receives the output of $p$ as input. We aim to train our objective network, $g_{\theta}^{\text{online}}$, by learning to predict the representations made by $g_{\xi}^{\text{target}}$, *i.e.*, knowledge distillation. The $f_{\xi}^b$ and $p_{\xi}$ networks have the same architecture as $f_{\theta}^b = f_{[l+1,\dots,L-1]}$ and $p_{\theta}$, respectively, but with different weight parameters $\xi$.

Since augmentations of input image space are not directly applicable in the feature space due to distribution shift, we propose *feature-level augmentation*. For a given feature vector $\mathbf{v}$, the transformed features $\mathbf{v}' = \text{Dropout}(\mathbf{v} + \epsilon', \gamma)$ and $\mathbf{v}'' = \text{Dropout}(\mathbf{v} + \epsilon'', \gamma)$, where $\epsilon', \epsilon'' \in \mathbb{R}^{H \times W \times C}$ sampled from $\mathcal{N}(\mu, \Sigma)$ are Gaussian noise vectors with the same shape as the feature $\mathbf{v}$, and $\gamma$ is the dropout ratio. By adding noise vectors, the feature $\mathbf{v}$ is augmented spatially because adjacent pixels of feature maps contain spatial information and are closely correlated. And, we apply the channel-wise dropout technique (Tompson et al., 2015), which means channel-wise sampling of feature maps. With augmented features $\mathbf{v}'$ and $\mathbf{v}''$, contrastive loss can be formulated as:

$$\mathcal{L}_{\theta,\xi}^{\text{contra}}(\mathbf{v}', \mathbf{v}'') := \left\| \frac{g_{\theta}(\mathbf{v}')}{||g_{\theta}(\mathbf{v}')||} - \frac{g_{\xi}(\mathbf{v}'')}{||g_{\xi}(\mathbf{v}'')||} \right\|^2, \tag{3}$$

where $|| \cdot ||$ denotes $L^2$-norm. For the real and fake features, we optimize the online network $g_{\theta}$ using the contrastive loss $\mathcal{L}_{\theta,\xi}^{\text{contra}}(\mathbf{v}_f', \mathbf{v}_f'') + \mathcal{L}_{\theta,\xi}^{\text{contra}}(\mathbf{v}_r', \mathbf{v}_r'')$. The target network $g_{\xi}$ is updated via the moving average technique expressed as $\tau\xi + (1 - \tau)\theta$, where $0 < \tau < 1$ denotes the increase in the target decay rate during training to adjust the weight updating ratio between $g_{\theta}^{\text{online}}$ and $g_{\xi}^{\text{target}}$. After

contrastive learning, only $f_\theta^b$ is used in conjunction with $f^a$ and $f_{[L]}$. Then, we train the classifier $f$ by cross-entropy loss with samples of the current task in supervised mode. The overall procedure for the proposed approach is presented in Algorithm 1.

---

**Algorithm 1:** LwP: Learning without Prejudices

---

**Inputs** : Datasets for T tasks $\{\mathcal{D}_t = (\mathcal{X}_t, \mathcal{Y}_t)\}_{t=1}^T$, classifier $f = \{f^a, f^b, f_{[L]}\}$, online network
$\quad\quad\quad g_\theta^{\text{online}}$, target network $g_\xi^{\text{target}}$, dropout ratio $\gamma$, target decay rate $\tau$, $P_{\mathbf{z}}, \mathcal{N}(\mu, \Sigma)$

1 **for** $t = 1$ **to** $T$ **do**
2    **while** *not converged* **do**
      // Train with GAN
3       $(\mathbf{x}, y) \sim (\mathcal{X}_t, \mathcal{Y}_t)$ and $\mathbf{z} \sim P_{\mathbf{z}}$       // sample real data and noise
4       $\mathbf{v}_r \leftarrow f^a(\mathbf{x})$ and $\mathbf{v}_f \leftarrow \mathcal{G}(\mathbf{z})$       // obtain real and fake features
5       Optimize $\mathcal{G}$ and $\mathcal{D}$ by (2) using $\mathbf{v}_r, \mathbf{v}_f$
      // Train with Contrastive Learning
6       **for** $\mathbf{v} \in \{\mathbf{v}_r, \mathbf{v}_f\}$ **do**
7         $\epsilon', \epsilon'' \sim \mathcal{N}(\mu, \Sigma)$       // sample noises for augmentations
8         $\mathbf{v}' \leftarrow \text{Dropout}(\mathbf{v} + \epsilon', \gamma)$       // augment the feature vector
9         $\mathbf{v}'' \leftarrow \text{Dropout}(\mathbf{v} + \epsilon'', \gamma)$       // augment the feature vector
10         Calculate $\mathcal{L}_{\theta,\xi}^{\text{contra}}(\mathbf{v}', \mathbf{v}'')$ in (3)       // calculate contrastive loss
11         $\theta \leftarrow \text{Optimizer}(\nabla_\theta \mathcal{L}_{\theta,\xi}^{\text{contra}}(\mathbf{v}', \mathbf{v}''), \theta)$    // optimize only $\theta$ and not $\xi$
12         $\xi \leftarrow \tau\xi + (1 - \tau)\theta$       // update $\xi$ by the moving average
13       **end**
      // Train with Classifier
14       Optimize classifier $f$ using $\text{CrossEntropy}(f(x), y)$
15    **end**
16 **end**
   **Output** : classifier $f$

---

## 5   Experiments

In this section, we experimentally evaluate the proposed method and compare it with several state-of-the-art models. We used three biased datasets: Biased MNIST (Bahng et al., 2020), Biased CIFAR-10 (Hendrycks & Dietterich, 2019), and Biased CelebA-HQ modified from (Karras et al., 2017).

### 5.1   Datasets

For **Biased MNIST**, we use the experimental setup in Section 3.1 to evaluate the model's generalizability. Following the bias planting protocol proposed by Nam et al. (2020), we create the **Biased CIFAR-10**. For the types of corruption $\{Snow, Frost, Fog, Brightness, Contrast, Spatter, Elastic, JPEG, Pixelate, Saturate\}$, we set each as a bias corresponding to the target object. The biases are changed for each task in exactly the same way as the Biased MNIST, indicating (airplane, snow), (automobile, frost), . . . , (truck, saturate) in the first task, and (airplane, frost), (automobile, fog), . . . , (truck, snow) in the second task. The unbiased test set exhibited a uniform distribution of corruption types along the target object. Thereby, the number of training image samples for each task is 4,000, and the test set has 10,000 images. We set $\beta = 0.85$ for Biased MNIST and Biased CIFAR-10. Among the attributes of images in CelebA-HQ (Karras et al., 2017), we set 'gender' as the target label and select 'makeup' and 'hair color' as the bias of the first and second task, respectively, because they have a significant correlation with 'gender' in the dataset. We name this sampled dataset as **Biased CelebA-HQ.** Thus, randomly sampled images for training are $\{(HeavyMakeup, Female), (NoMakeup, Male)\}$ and $\{(BlondHair, Female), (BlackHair, Male)\}$ for each task, respectively. We additionally set the bias of the third task by 'hair length' utilizing the public annotation provided by Jeon et al. (2022); hence, the training set consists of $\{(LongHair, Female), (ShortHair, Male)\}$ images. All pairs of $(attribute, gender)$ for training have 2,000 samples, and the unbiased test set is composed of 100 images to be evenly distributed. For CelebA-HQ, we do not split biased data for validation because there are not enough biased pair samples. The detailed distribution of the training and test sets is presented in the Appendix.

Table 1: Average accuracy $\overline{Acc}$ and average unbiased accuracy $\overline{Acc_{ub}}$ all along the datasets. `Base` denotes simple CNN for Biased MNIST and ResNet-18 for Biased CIFAR-10, and Biased CelebA-HQ without any additional regularization for forgetting or unbiasing. Replay (200) means replay-based approaches with 200 previous samples buffer. We display the best performance by **bold** and second performance by underline. All experiments run on three different random seeds and we report the means and standard deviations.

| Dataset | | MNIST | | CIFAR-10 | | CelebA-HQ |
|---|---|---|---|---|---|---|
| | | $\overline{Acc_{ub}}$ | $\overline{Acc}$ | $\overline{Acc_{ub}}$ | $\overline{Acc}$ | $\overline{Acc_{ub}}$ |
| *Continual Unbiased Learning* | | | | | | |
| | Base | 84.66($\pm$2.41) | 89.22($\pm$6.05) | 26.58($\pm$0.45) | 30.98($\pm$0.41) | 74.67($\pm$1.15) |
| **Regularization** | EWC | 68.32($\pm$4.35) | 67.56($\pm$2.59) | 27.03($\pm$0.68) | 31.73($\pm$1.17) | 77.33($\pm$1.45) |
| | LwF | 83.58($\pm$0.71) | 89.95($\pm$1.80) | 25.24($\pm$0.57) | 27.16($\pm$1.51) | 74.56($\pm$0.51) |
| **Generator** | DGR | 85.04($\pm$1.32) | 92.77($\pm$0.16) | 28.94($\pm$2.61) | 32.08($\pm$7.00) | 71.56($\pm$2.87) |
| | GFR | 77.50($\pm$2.78) | 78.49($\pm$1.90) | 29.11($\pm$0.48) | 33.95($\pm$1.00) | 64.78($\pm$4.07) |
| | ABD | 82.78($\pm$1.05) | 87.44($\pm$0.72) | 29.67($\pm$0.61) | 34.40($\pm$0.89) | 79.11($\pm$4.17) |
| **Unbiasing** | LfF | 82.48($\pm$2.45) | 86.85($\pm$3.89) | 25.19($\pm$0.77) | 27.50($\pm$2.52) | 72.44($\pm$3.34) |
| | LwP (w/o buffer) | **91.57($\pm$0.82)** | **95.58($\pm$0.46)** | **31.18($\pm$0.29)** | **35.16($\pm$2.88)** | **79.78($\pm$1.90)** |
| *Continual Unbiased Learning with Replay Buffer* | | | | | | |
| **Replay (200)** | HAL | 82.40($\pm$2.21) | 77.08($\pm$9.32) | 23.50($\pm$8.00) | 26.48($\pm$14.26) | 74.56($\pm$4.86) |
| | DER | 90.37($\pm$1.33) | 95.05($\pm$0.29) | 31.07($\pm$0.53) | **38.42($\pm$0.64)** | 75.00($\pm$6.77) |
| | LiDER | 89.44($\pm$6.07) | **96.88($\pm$0.43)** | 28.02($\pm$0.48) | 30.89($\pm$0.94) | 77.44($\pm$2.34) |
| | LwP (w/ buffer) | **92.34($\pm$0.55)** | 96.02($\pm$0.55) | **32.06($\pm$0.59)** | 30.99($\pm$1.38) | **80.89($\pm$1.68)** |

## 5.2 EXPERIMENTAL SETUP

**Competing models.** Previous approaches to CL can be categorized into regularization, replay, and generator-based approaches. We set all the categories on a competing line. We add an unbiasing method, `LfF` (Nam et al., 2020), although it is not designed for CL, it can be applied for CUL. Thus, we compare the proposed method with `EWC` (Kirkpatrick et al., 2017), `LwF` (Li & Hoiem, 2017), `DGR` (Shin et al., 2017), `GFR` (Liu et al., 2020), `ABD` (Smith et al., 2021), `HAL` (Chaudhry et al., 2021), `DER` (Buzzega et al., 2020), `LiDER` (Bonicelli et al., 2022) and `LfF` for all the datasets. We do not consider previous samples during training for a new task in our model design. Because it cannot be deployed when access to prior data is strictly limited due to privacy problems, *e.g.*, personal medical or credit information. Nonetheless, for a fair comparison to the replay-based method, we evaluate our method with a buffer. We set the buffer size as 200.

**Implementation details.** For the Biased MNIST, we employ a simple CNN composed of four convolutional layers and three fully connected layers as the baseline model and backbone network of all competing models. For Biased CIFAR-10 and Biased CelebA-HQ, ResNet-18 (He et al., 2016) is used as the backbone. For each experiment, we used the Adam optimizer (Kingma & Ba, 2014), grid search learning rate (initial value and decay schedule), stopping criterion, and batch size. For hyperparameters in competing models, we follow the same setting as that presented in this paper. However, as we experimentally found that the learning rate is important in CUL, we fine-tune it and display the result if better than the original setting. More implementation details are provided in the Appendix. We set $l$ by experiments reported in the Appendix.

## 5.3 EXPERIMENTAL RESULTS

Table 1 exhibits the evaluation results of `LwP` and state-of-the-art models on CUL. From the experiments on three intentionally biased datasets, we find that regularization-based models are limited to CUL, showing performance degradation from the `Base` model on some datasets. It is notable that the generator-based methods show remarkable generalizability for some experiments. This implies that generating prior samples and feeding them contribute to CUL. Nonetheless, utilizing only the generated prior samples is limited to encouraging *benign forgetting* (experiments on Biased CelebA-HQ). Based on the generator-based approach, `LwP` significantly increase performance with *contrastive learning*. This quantitatively demonstrates *contrastive learning* contributes for *benign forgetting* (It is also demonstrated in Table 2). Further, it is noteworthy that `LwP` (w/o buffer) outperforms

replay-based methods on some datasets (Biased MNIST, Biased CelebA-HQ). The unbiased learning method, `LfF`, deteriorates the performance for all the datasets, exhibiting performance degradation compared to `Base`.

## 5.4 ANALYSIS

Table 2: Ablation study on components of `LwP`. We use Biased MNIST for evaluation. The model with none of them applied is the base model, simple CNN.

| SSL | | ✓ | ✓ |
|---|---|---|---|
| GAN | | | ✓ |
| $\overline{Acc_{ub}}$ | 84.66($\pm$2.41) | 89.79($\pm$1.26) | **91.57($\pm$0.82)** |

Table 3: Ablation study on self-supervised learning. We use Biased MNIST for evaluation. All the architecture and experimental setup except for SSL are exactly the same.

| SSL | MoCo | DINO | Barlow Twins | BYOL (Ours) |
|---|---|---|---|---|
| $\overline{Acc_{ub}}$ | 83.86($\pm$0.71) | 84.23($\pm$0.39) | 86.66($\pm$0.66) | **91.57($\pm$0.82)** |

**Analysis of `LwP` components.**  We conduct an ablation study to evaluate the contribution of each presented component in section 4. Table 2 shows that the self-supervised technique (BYOL) and feature generator contribute to the generalizability of the model.

**Choice of self-supervised learning.**  We compare the performance of the models using different self-supervised learning (SSL) approaches (He et al., 2020; Grill et al., 2020; Caron et al., 2021; Zbontar et al., 2021). MoCo (He et al., 2020) requires large memory banks with contrastive loss, called infoNCE (Oord et al., 2018). BYOL (Grill et al., 2020) proposed a metric learning approach trained using a momentum encoder. DINO (Caron et al., 2021) complemented BYOL with similarity matching loss and mean-teacher self-distillation (Tarvainen & Valpola, 2017). Barlow Twins (Zbontar et al., 2021) proposed an objective function that measures the cross-correlation matrix between two embeddings via an identical network from two different distorted samples. MoCo is limited in its use because of the unacceptable computational costs of maintaining ($positive, negative$) pairs. We decide on BYOL with our SSL technique through an experimental comparison between BYOL, DINO, and Barlow Twins. Table 3 shows that the BYOL method outperforms other SSL methods.

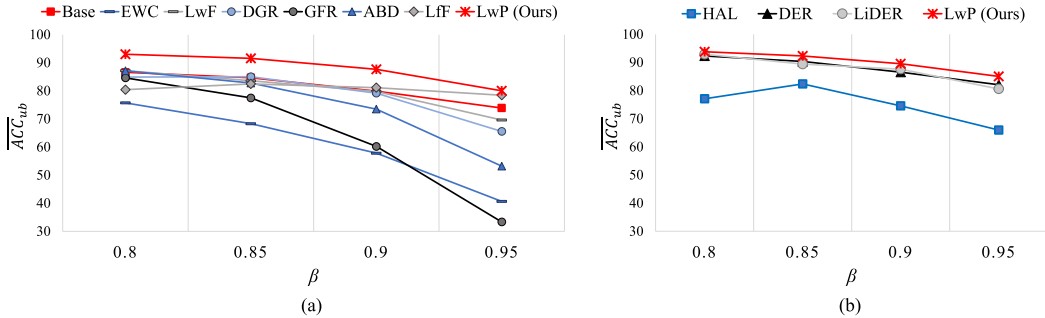

(a)                                                                  (b)

Figure 3: Evaluations on various $\beta$. (a) Continual unbiased learning. (b) Continual unbiased learning with replay buffer. We evaluate all the experiments with Biased MNIST.

**Generalization of `LwP` on various $\beta$.**  In reality, the training dataset is biased to varying degrees. Thus, we evaluate `LwP` on several $\beta$ and compare it to state-of-the-art methods. In Figure 3, `LwP` exhibits the best generalizability for all the $\beta$ in both cases.

## 6 RELATED WORK

### 6.1 CONTINUAL LEARNING

Recent literature on continual learning can be categorized into regularization-based, replay-based, and generator-based methods.

**Regularization-based methods.**  `EWC` (Kirkpatrick et al., 2017) approximated the importance of parameters from a probabilistic perspective and regularized the update of decisive ones training a new task. `LwF` (Li & Hoiem, 2017) has multiple task-specific heads. It records probabilities obtained from the previous task heads and uses them as targets of surrogate loss when learning a new task. Chaudhry et al. (2018) regularized parameter updating in a current task such that the new conditional likelihood is close to the previously learned one in terms of KL-divergence. Aljundi et al. (2018)

presented memory-aware synapses (MAS) that estimate the importance of the parameters in a model and then penalize changing them in new tasks. In further work, Aljundi et al. (2019a) investigated how to transform MAS into an online setup where the data distribution changes and the tasks are not specified. Dhar et al. (2019) presented an attention-based approach that incrementally learns new classes by restricting divergence. Ahn et al. (2019) presented an uncertainty-regularized continual learning framework based on Bayesian online learning. Douillard et al. (2020) proposed spatial-based distillation loss applied throughout the model.

**Replay-based methods.** Inspired by the first replay-based approach for catastrophic forgetting and experience replay (Robins, 1995), the majority of related studies have been devoted to replay-based methods. Lopez-Paz & Ranzato (2017) united rehearsal methods with knowledge distillation and regularization, making the model sufficiently close to the previous model. Aljundi et al. (2019b) considered a sampling of prior tasks as a constraint reduction problem, resulting in maximizing the diversity of samples. Mai et al. (2021) suggested that softmax classifiers could cause recency bias in continual learning and hence exploited the nearest-class-mean classifier, instead. For preferable clustering, data samples from previous tasks are saved and utilized in the current training step. Lin et al. (2021) suggested continual contrastive self-supervised learning via a rehearsal method, which preserves the feature vectors using k-means clustering from the previous dataset. Madaan et al. (2021) presented a technique interpolating previous and current instances. DER (Buzzega et al., 2020) exploited logits sampled during the optimization trajectory to encourage consistency with its past. HAL (Chaudhry et al., 2021) complemented a new objective termed anchoring, where the model is bi-level optimized. LiDER (Bonicelli et al., 2022), constrained the backbone network by its layer-wise Lipschitz constants with respect to replay samples.

**Generator-based methods.** Although the replay-based method is an intuitive approach for tackling catastrophic forgetting, it cannot be deployed if there is a privacy issue regarding data. As an alternative, generator-based methods that do not directly store prior samples were presented. DGR (Shin et al., 2017) obtained past information via a GAN, making pseudo-labels from the discriminator. Kemker & Kanan (2017) generated prior informative images via auto-encoder, where the encoder approximates pseudo labels, and the decoder makes images guided by reconstruction loss. Xiang et al. (2019) exploited a GAN conditioned on the labeled features embedded by the discriminator, which allows explicit supervised learning for the classifier. Ostapenko et al. (2019) presented dynamic generative memory that employs an auxiliary classifier GAN with an increased number of parameters. In each task, binary masks were applied to concentrate on influential parameters. GFR (Liu et al., 2020) proposed a feature generator, reducing the complexity of generative replay and preventing the imbalance problem. Yin et al. (2020) and ABD (Smith et al., 2021) applied 'inversion', which makes class-conditional input images from random noise via a trained network.

## 6.2 Unbiased Learning

Unbiased learning is a branch of robustness in machine learning. As machine-learning algorithms are overly dependent on the distribution of training datasets, models are often biased, causing unreliable generalization at inference (Torralba & Efros, 2011). A line of recent works has been dedicated to mitigating this issue. Most of the studies assumed that the biases in datasets are known (*e.g.*, color, texture) and exploited this information by designing various models (Kim et al., 2019; Geirhos et al., 2019; Bahng et al., 2020; Gong et al., 2020; Adeli et al., 2021; Dhar et al., 2021). However, with the motivation that a mechanism based on bias predefined by human knowledge is unsuited for image datasets including countless sensory attributes, LfF (Nam et al., 2020) and Jeon et al. (2022) addressed the challenge of unknown biases.

## 7 Conclusion and Future Work

A large body of literature suggests that forgetting while learning a sequence of tasks is catastrophic. However, in this study, we found that forgetting could encourage the generalizability of a model if the dataset has bias. Based on this motivation, our proposed method, LwP, encourages benign forgetting and regularizes malignant forgetting for continual unbiased learning via a *feature generator* and *contrastive learning* in conjunction with *feature-level augmentation*. Experimentally, the LwP contributes to generalization, while conventional CL methods are limited to unbiasing. In terms of further work, extensive exploration of the relationship between forgetting and various incomplete data distributions, *e.g.*, imbalanced or mislabeled data distributions, are potentially value-adding future research directions.

**Acknowledgement.** This work was supported by the NRF grant [2012R1A2C3010887] and the MSIT/IITP([1711117093], [2021-0-00077], [No. 2021-0-01343, Artificial Intelligence Graduate School Program (SNU)]).

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

## A    EXPLORATION ON THE DATA DISTRIBUTION

We construct the biased training set and unbiased test set from Background Colored MNIST, Corrupted CIFAR-10, and CelebA-HQ to set up continual unbiased learning. Figure 4, 5, 6, 7 exhibit the overall distribution of each task on each dataset. And, the examples of Biased CIFAR-10 and Biased CelebA are displayed in Figure 8.

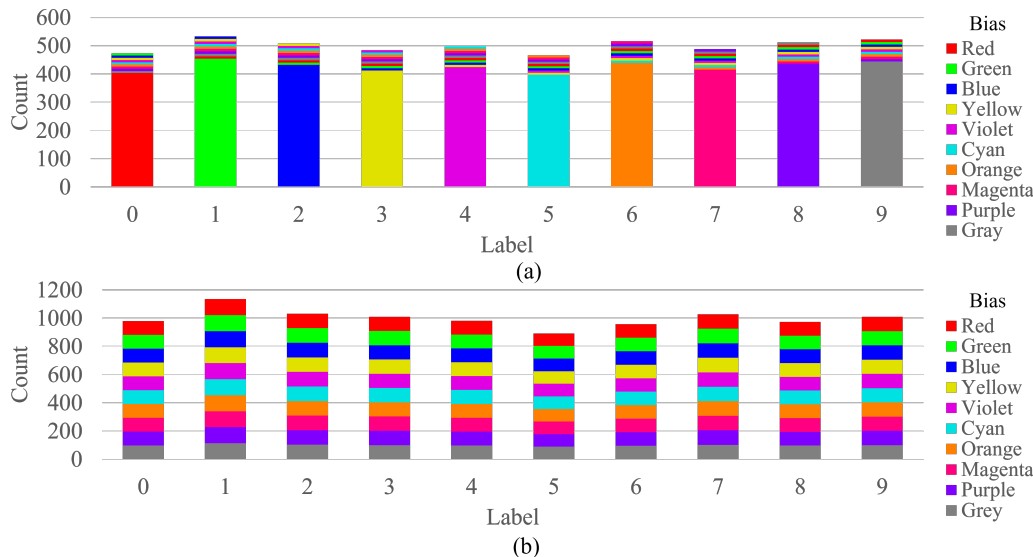

Figure 4: **Distribution of Biased MNIST.** (a) Training set ($\beta = 0.85$). We randomly sampled images from MNIST to make ten subsets allocating them as training sets for ten tasks. Following the same sampling scenario, we just slid bias (color) whenever the task is changed from 1 to 10. (b) Test set. The colors are uniformly distributed for each label, denoted by an unbiased set.

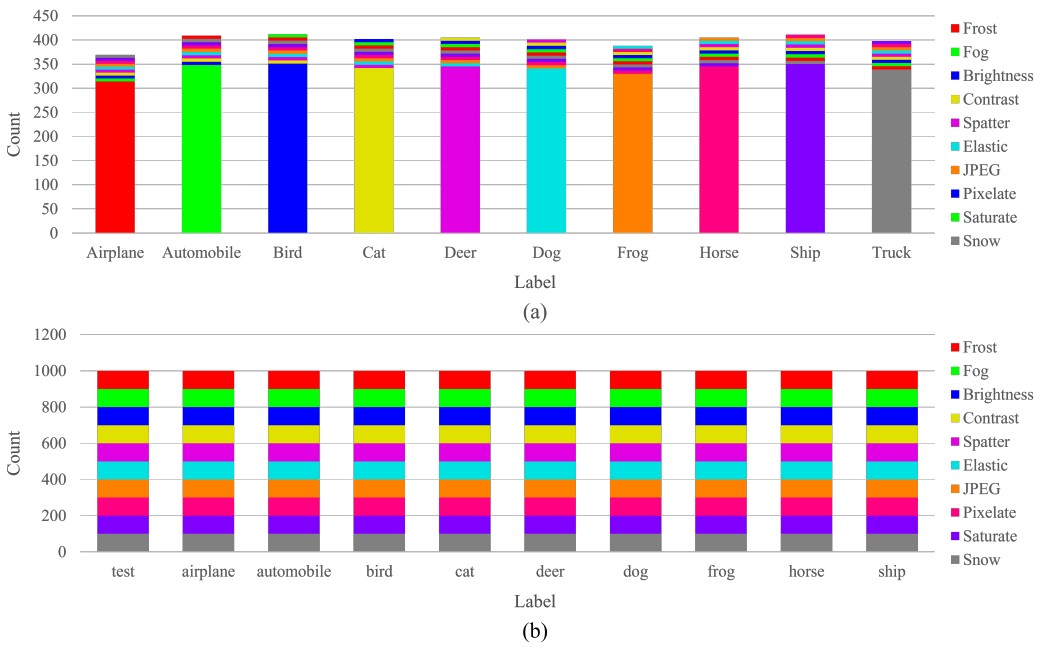

Figure 5: **Distribution of Biased CIFAR-10.** (a) Training set ($\beta = 0.85$). We construct datasets in exactly the same way as Biased MNIST. (b) Unbiased test set.

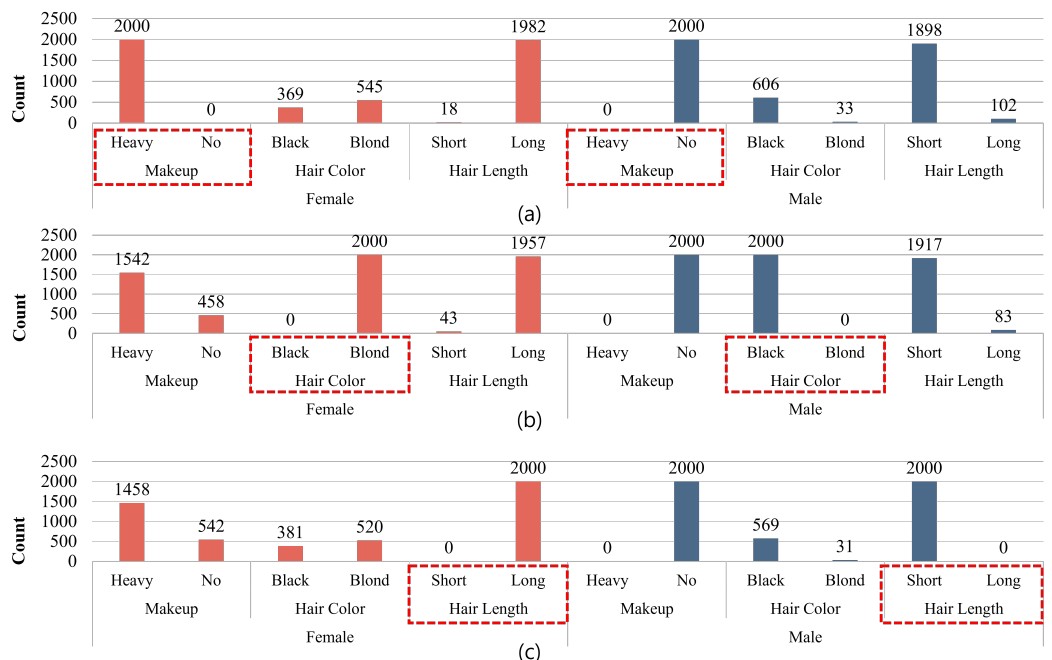

Figure 6: **Distribution of Biased CelebA-HQ training set.** (a) Task1. 2,000 (Female, HeavyMakeup) and 2,000 (Male, NoMakeup) images are randomly sampled. (b) Task2. 2,000 (Female, BloncHair) and 2,000 (Male, BlackHair) images. (c) Task3. 2,000 (Female, LongHair) and 2,000 (Male, ShortHair) images. However, as a face image include several attributes, we display all these pairs.

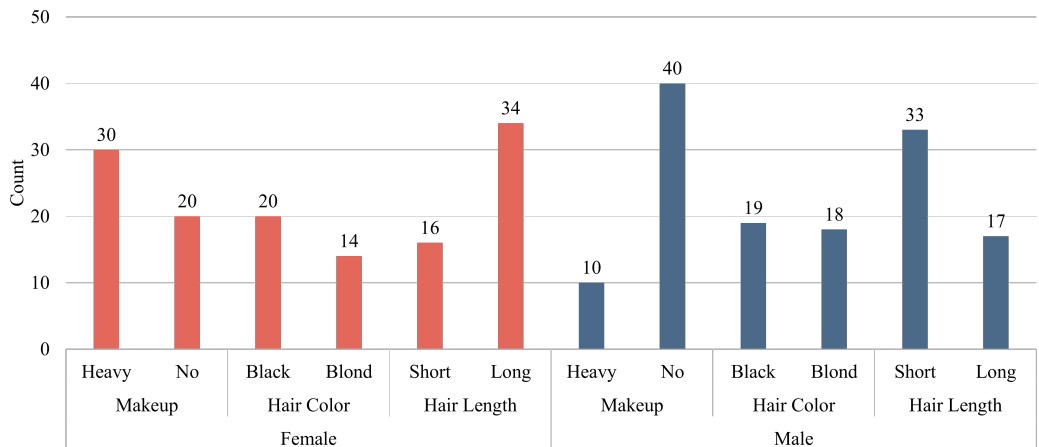

Figure 7: **Distribution of CelebA-HQ test set.** We randomly sampled test set to be uniform for all the pairs. However, as the limited number of image samples, *e.g.*, (Male, HeavyMakeup), and such samples are biased, *e.g.*, all the (Male, HeavyMakeup) have short hair, the distribution could not be strictly uniform.

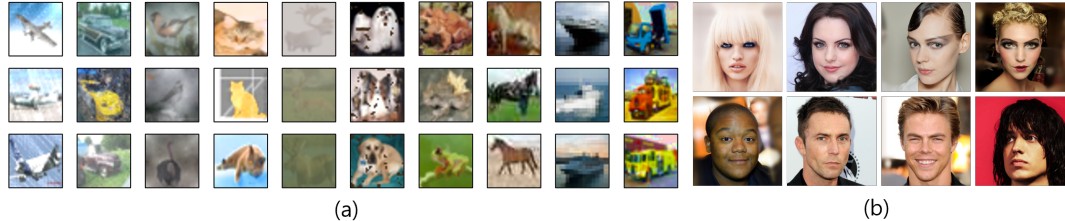

(a)                    (b)

Figure 8: Experimental dataset. (a) Biased CIFAR-10. (b) Biased CelebA-HQ. We display image samples of the first task. For CIFAR-10, corruption is the bias attribute, *e.g.*, all the airplanes are corrupted by snow noise. Each time the task changes, the bias slides one by one. For CelebA-HQ, all the women put on makeup, while the men do not. Hair color and hair length are bias for the second and third tasks, respectively.

## B  EXPERIMENT DETAILS

### B.1  ARCHITECTURE

**Biased MNIST.** The architecture of the classifier $f$ for the Biased MNIST is composed of four convolutional (CONV) layers, a global average pooling (GAP) layer, and 3 fully-connected (FC) layers (please see Table 4). In this paper, We set $l = 2$ splitting $f$ into $\{f_{[1,2]}, f_{[3,4,5,6,7]}, f_{[8]}\}$. Hence, $f^a$ and $f^b$ are $f_{[1,2]}$ and $f_{[3,4,5,6,7]}$, respectively. $g_\theta^{\text{online}} = \{f_\theta^b, p_\theta, q_\theta\}$ and $g_\xi^{\text{target}} = \{f_\xi^b, p_\xi\}$ are used for *contrastive learning*, where $p$ and $q$ are FC layers with 128 units. Since the generator, $\mathcal{G}$ makes fake features with the same shape as the output of encoder $f^a$, the generated features $\mathbf{v}_f$ via $\mathcal{G}$ belong to $\mathbb{R}^{28 \times 28 \times 32}$ as feature maps activated from the input image by 4 CONV layers. Note that $l$ denotes an $l$-th layer, *e.g.*, $l = 5$ means GAP layer and $l = 7$ is the second FC layer.

Table 4: **Classifier $f$, Simple CNN.** CONV_n denotes the CONV layer and FC_n denotes the FC layer for Simple CNN. $H$ and $W$ mean the height and width of input images.

| $l$ | Layer Name | Type | Output Size |
|---|---|---|---|
| 1 | CONV_1 | $[3 \times 3, 16]$ | $H \times W \times 16$ |
| 2 | CONV_2 | $[3 \times 3, 32]$ | $H \times W \times 32$ |
| 3 | CONV_3 | $[3 \times 3, 64]$ | $H \times W \times 64$ |
| 4 | CONV_4 | $[3 \times 3, 128]$ | $H \times W \times 128$ |
| 5 | GAP | · | 128 |
| 6 | FC_1 | 128 (units) | 128 |
| 7 | FC_2 | 128 (units) | 128 |
| 8 | FC_3 | 10 (units) | 10 |

**Biased CIFAR-10 and Biased CelebA-HQ.** For Biased CIFAR-10 and Biased CelebA-HQ, we use ResNet-18 as our classifier $f$ as depicted in table 5. We modify the first CONV of ResNet-18 with the kernel size $3 \times 3$, instead of $7 \times 7$. For convenience, we consider a block as a layer $l$. We split $f$ into $\{f_{[1]}, f_{[2,3,4,5]}, f_{[6]}\}$ resulting in $f^a = f_{[1]}$ and $f^b = f_{[2,3,4,5]}$ for Biased CIFAR-10 with $l = 1$ and set $l = 2$ for Biased CelebA-HQ. We choose $l$ by ablation study. We set $g_\theta^{\text{online}} = \{f_\theta^b, p_\theta, q_\theta\}$ and $g_\xi^{\text{target}} = \{f_\xi^b, p_\xi\}$, where $p$ and $q$ are FC layers with 512 units. Thus, the size of $v_f$ generated by $\mathcal{G}$ is $H \times W \times 64$ for Biased CIFAR-10 and $H/2 \times W/2 \times 128$ for Biased CelebA-HQ, where $H$ and $W$ denotes the height and width of input images.

**Generator $\mathcal{G}$ and discriminator $\mathcal{D}$.** We set the architecture of the *feature generator* $\mathcal{G}$ by one FC layer and three CONV layers. During generating features, we increase the size of feature maps in $\mathcal{G}$ with two interpolation layers. The discriminator $\mathcal{D}$ consists of four CONV layers and an FC layer to distinguish whether input features are real or fake. The leaky ReLU activation layer follows every CONV layer. We adjusted the number of units in the FC layer for $\mathcal{G}$ to fit the different shapes of the target feature maps for the three datasets while maintaining the overall architecture.

Table 5: **Classifier $f$, ResNet-18.** Block_n denotes the basic building block for ResNet. $H$ and $W$ mean the height and width of input images.

| $l$ | Layer Name | Type | Output Size |
|---|---|---|---|
| | | $[\,3 \times 3,\, 64\,]$ | |
| 1 | Block_1 | $\begin{bmatrix} 3 \times 3,\ 128 \\ 3 \times 3,\ 128 \end{bmatrix} \times 2$ | $H \times W \times 64$ |
| 2 | Block_2 | $\begin{bmatrix} 3 \times 3,\ 128 \\ 3 \times 3,\ 128 \end{bmatrix} \times 2$ | $H/2 \times W/2 \times 128$ |
| 3 | Block_3 | $\begin{bmatrix} 3 \times 3,\ 256 \\ 3 \times 3,\ 256 \end{bmatrix} \times 2$ | $H/4 \times W/4 \times 256$ |
| 4 | Block_4 | $\begin{bmatrix} 3 \times 3,\ 512 \\ 3 \times 3,\ 512 \end{bmatrix} \times 2$ | $H/8 \times W/8 \times 512$ |
| 5 | GAP | . | 512 |
| 6 | FC | # of classes (units) | # of classes |

## B.2 TRAINING PROCEDURE

To train the classifier $f$, for both supervised learning with samples of the current task and contrastive learning with previous samples, we use Adam optimizer with learning rate $10^{-4}$, weight decay $5 \times 10^{-4}$, and $(\beta_1, \beta_2) = (0.9, 0.999)$. To train the generator $\mathcal{G}$ and discriminator $\mathcal{D}$, we use Adam optimizers with learning rate $5 \times 10^{-5}$ for $\mathcal{G}$ and $2 \times 10^{-4}$ for $\mathcal{D}$. We set $(\beta_1, \beta_2) = (0.5, 0.999)$ for Adam in $\mathcal{G}$ and $\mathcal{D}$. To optimize the generator $\mathcal{G}$ and discriminator $\mathcal{D}$, we use the WGAN-GP. We set input image size as $28 \times 28$ for Biased MNIST, $32 \times 32$ for Biased CIFAR-10, and $128 \times 128$ for Biased CelebA-HQ. The batch size and epochs per task are 32 and 20, respectively for all the experiments.

For *feature-level augmentation*, we set channel-wise applied dropout rate $\gamma = 0.2$, and use $\mu = \mathbf{0}$ and $\Sigma = 0.005 * \mathbf{I}$ for $\mathcal{N}(\mu, \Sigma)$ when spatially augmenting. Following exactly the same setup of BYOL, we set variable $\tau := 1 - (1 - 0.996)(\cos(\pi k / K) + 1)/2$, where $k$ denotes the current training epoch and $K$ denotes the total number of training epochs.

## C EXPLORATION ON LwP

### C.1 INVESTIGATION FOR ARCHITECTURAL PARAMETER, $l$

Splitting layer $l$ is an important parameter in our model design. From the *feature generator* to online and target networks, all the architectures of the networks are decided by the parameter $l$. Therefore, we conduct a comparing experiment for $l$. By experiment in Table 6 and Table 7, we found all the $l$ show competitive generalizability compared to state-of-the-art CL methods. Among them, we chose the best one, $l = 2$, for our model. We conducted the same ablation experiments for Biased CIFAR-10 to choose $l$.

Table 6: **Comparison for splitting layer $l$.** We evaluate the models on Biased MNIST three times and display the average of them with standard deviation.

| $l$ | 1 | 3 | 4 | 2 (ours) |
|---|---|---|---|---|
| $\overline{Acc_{ub}}$ | 90.52($\pm$0.15) | 91.08($\pm$1.24) | 90.24($\pm$0.90) | **91.57($\pm$0.82)** |

Table 7: **Comparison for splitting layer $l$.** We evaluate the models on Biased CIFAR-10 three times and display the average of them with standard deviation.

| $l$ | 2 | 3 | 4 | 1 (ours) |
|---|---|---|---|---|
| $\overline{Acc_{ub}}$ | 29.79($\pm$0.53) | 26.39($\pm$0.54) | 25.59($\pm$0.87) | **31.18($\pm$0.29)** |

## C.2 Contribution of Feature Generator

Although several conventional CL approaches employed image generators to memorize previous samples, they are only 3-channel aggregated features. Thus, if the generator does not work well, generated input image could disturb the desirable optimization of the classifier. Instead, generated features can provide more information to be used selectively and hence make the classifier more robust. Table 8 experimentally demonstrates our intuition.

Table 8: **Comparison for image and feature generation.** We evaluate all the models on Biased MNIST three times and display the average of them with standard deviation.

| | Dataset | Image | Feature (ours) |
|---|---|---|---|
| $\overline{Acc_{ub}}$ | MNIST | 88.80($\pm$1.36) | **91.57($\pm$0.82)** |
| $\overline{Acc_{ub}}$ | CIFAR-10 | 25.33($\pm$0.74) | **31.18($\pm$0.29)** |

**Choice of self-supervised learning.** We compare the performance of the models using different self-supervised learning (SSL) approaches MoCo, DINO, Barlow Twins, and BYOL for Biased CIFAR-10 following the exactly same way as Biased MNIST.

Table 9: Ablation study on components of `LwP`. We use Biased CIFAR-10 for evaluation. The model with none of them applied is the base model, ResNet-18.

| SSL | | ✓ | ✓ |
|---|---|---|---|
| GAN | | | ✓ |
| $\overline{Acc_{ub}}$ | 26.58($\pm$0.45) | 30.36($\pm$0.29) | **31.18($\pm$0.29)** |

Table 10: Ablation study on self-supervised learning. We use Biased CIFAR-10 for evaluation. All the architecture and experimental setup except for SSL are exactly the same.

| SSL | MoCo | DINO | Barlow Twins | BYOL (Ours) |
|---|---|---|---|---|
| $\overline{Acc_{ub}}$ | 31.00($\pm$0.57) | 27.81($\pm$0.63) | 30.61($\pm$0.36) | **31.18($\pm$0.29)** |

## C.3 Model size

Generally, the number of parameters affects the performance of the model. Although simple comparison for the number of parameters is not fair because many methods for CL address malignant forgetting by regularization-based (additional operation for regularization) and replay-based (more training iterations with buffer) approaches, which are not related to model size. Nonetheless, generator-based methods can be compared for model size. Therefore, we compare the model size and present a tiny version of `LwP` in Table 11. It is notable that `LwP(tiny)` also exhibits competitive performance.

## C.4 Replay Buffer

Learning with replay buffer can help the model to memorize previous information, discouraging *malignant forgetting*. Although there are some special cases, where access to prior samples is strictly limited (*e.g.*, privacy data such as medical or financial data, with a short storage period), a replay buffer is often available. Therefore, we compare the generalizability of `LwP` and replay-based methods with small-sized (200) and big-sized buffers (2,000). As a result, Table 12 shows that our method achieves the best performance in both scenarios.

Table 11: Model size. We investigate the number of parameters for generator-based methods. Since we set the classifier for all the models as ResNet18, we compare GAN models, generator $\mathcal{G}$, and $\mathcal{D}$.

| | Dataset | DGR | GFR | ABD | LwP | LwP(tiny) |
|---|---|---|---|---|---|---|
| $\mathcal{G}$ | | 2.30 M | 0.44 M | 6.50 M | 6.52 M | 0.38 M |
| $\mathcal{D}$ | MNIST | 2.76 M | 0.34 M | - | 2.77 M | 0.18 M |
| Total | | 5.06 M | 0.78 M | 6.50 M | 9.29 M | 0.56 M |
| $\overline{Acc_{ub}}$ | | 85.04($\pm$1.32) | 77.50($\pm$2.78) | 82.78($\pm$1.05) | 91.57($\pm$0.82) | 91.33($\pm$0.63) |
| G | | 3.58 M | 0.63 M | 8.42 M | 8.46 M | 2.07 M |
| D | CIFAR-10 | 2.76 M | 0.54 M | - | 2.82 M | 0.72 M |
| Total | | 6.34 M | 1.17 M | 8.42 M | 11.28 M | 2.79 M |
| $\overline{Acc_{ub}}$ | | 28.94($\pm$2.61) | 29.11($\pm$0.48) | 29.67($\pm$0.61) | 31.18($\pm$0.29) | 30.65($\pm$1.73) |

Table 12: Buffer size. With small (200) and big (2,000) buffers, we estimate $\overline{Acc_{ub}}$ and report them.

| | MNIST | | CIFAR-10 | |
|---|---|---|---|---|
| | 200 | 2,000 | 200 | 2,000 |
| HAL | 82.40($\pm$2.21) | 84.28($\pm$0.75) | 23.50($\pm$8.00) | 29.24($\pm$0.48) |
| DER | 90.37($\pm$1.33) | 89.84($\pm$0.72) | 31.07($\pm$0.53) | 34.15($\pm$0.21) |
| LiDER | 89.44($\pm$6.07) | 92.61($\pm$2.14) | 28.02($\pm$0.48) | 35.72($\pm$0.76) |
| LwP | **92.34($\pm$0.55)** | **93.14($\pm$0.95)** | **32.06($\pm$0.59)** | **40.18($\pm$0.37)** |

## C.5 INVESTIGATION ON MORE VARIOUS $\beta$

Collecting data from the real world, the degree of bias $\beta$ could be various. To investigate the generalizability of `LwP` and competing models, we set $\beta$ $0.1 \sim 0.8$ at intervals of 0.1 along with $0.8 \sim 0.95$ at intervals of 0.05. Figure C.5 shows that our method generalizes best for all $\beta$.

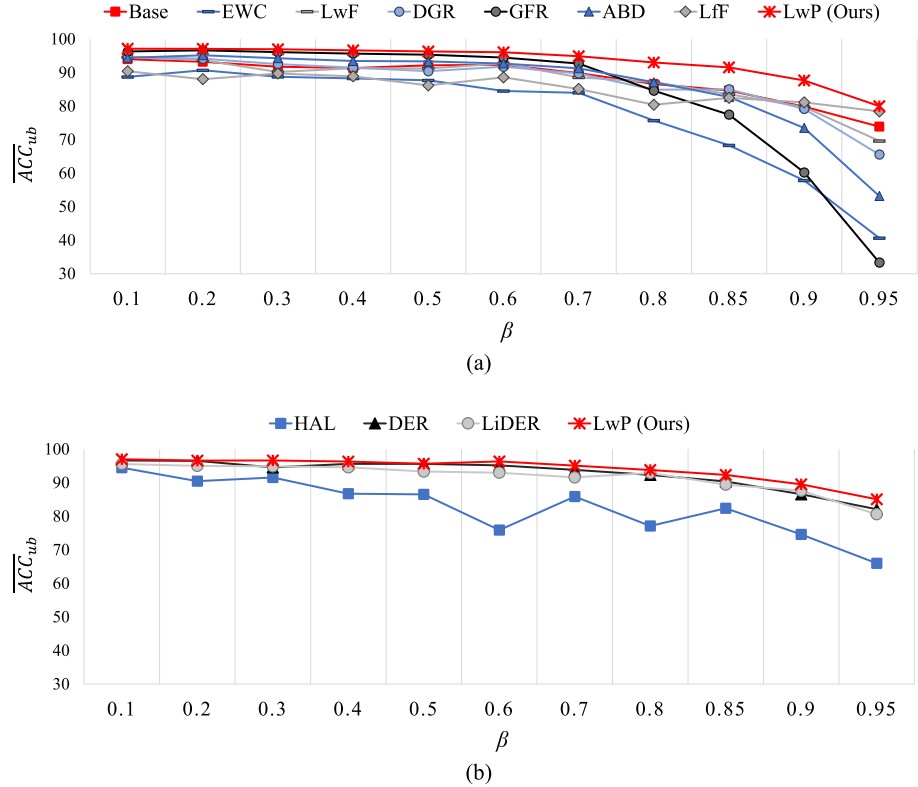

Figure 9: Evaluations on various $\beta$. (a) Continual unbiased learning. (b) Continual unbiased learning with replay buffer. We evaluate all the experiments with Biased MNIST.

