# OpenReview forum: "Learning without Prejudices: Continual Unbiased Learning via Benign and Malignant Forgetting"
_ICLR.cc/2023/Conference — ICLR 2023 poster_

### Official Review · Reviewer_ZsCC · 2022-10-23

**Confidence:** 5
**Correctness:** 3
**Technical Novelty And Significance:** 3
**Empirical Novelty And Significance:** 3
**Recommendation:** 8

**Clarity, Quality, Novelty And Reproducibility:**

Good quality.
Good clarity.
Nice originality.

**Strength And Weaknesses:**

Strengths:

(a) This paper is well-written and easy to read.

(b) The phenomenon found in this work is easy to follow.

(c) I appreciate the method proposed in this paper for its motivations and great performance in extensive experiments.

Weaknesses:

(a) Please describe BYOL in details.

(b) The proposed method splits the classifier into $[f^a,f^b,f_{{L}}]$. Why training $f^a$ by GANg can address malignant forgetting? Why training $f^b$ by BYOL contrastive learning can encourage benign forgetting?

(c) In the final step of Algorithm 1, $f^a$ and $f^b$ will be retrained?

(d) In practice, how to determine $l$ and $L$ in Section 4.2?

**Summary Of The Paper:**

This paper focuses on the bias in continual learning. According to the bias, this paper designs a novel method, named learning without prejudices (LwP) to discourage malignant forgetting and encourage benign forgetting. The main contributions can be summarized as:

(a) Presenting a novel framework, termed "continual unbiased learning", continual unbiased learning benchmarks and an evaluation protocol.

(b) Categorizing the forgetting into malignant forgetting and benign forgetting.

(c) Proposing a novel method, Learning without Prejudices (LwP).

**Summary Of The Review:**

I recommend accept, good paper.

---

> ### Author Response · Authors · 2022-11-17
> **Thanks for valuable advice.**
>
> > Please describe BYOL in detail.
>
> We are really sorry for missing the detailed presentation for BYOL. We add it in the Appendix 'Description of BYOL'.
>
> > The proposed method splits the classifier into [$f^a, f^b, f_L$]. Why training $f^a$ by GANs can address malignant forgetting? Why training $f^b$ by BYOL contrastive learning can encourage benign forgetting?
>
> Thanks for this comment. This advice gave us an opportunity to reconsider our presentation for *discouraging malignant forgetting, encouraging benign forgetting*. Firstly, let us revisit the overall procedure depicted in Algorithm 1.
>
>
> - training generator $\mathcal{G}$ to generate feature maps $v_f$ that includes previous information. Discriminator $\mathcal{D}$ receives $\mathbf{v}_f = \mathcal{G}(\mathbf{x})$ and $\mathbf{v}_r = f^a(\mathbf{x})$ to regularizes $\mathcal{G}$ to make real-like features.
> - training the network $f^b$ with $\{\mathbf{v}_r, \mathbf{v}_f\}$ by contrastive learning.
> - training $f = \\{f^a, f^b, f_{[L]}\\}$ with the current real images $\mathbf{x}$ by cross-entropy.
>
> Then, let us present intuitions on this algorithm. 1) discouraging malignant forgetting is to memorize previous information. Thus, generating past feature maps $v_f$ by $\mathcal{G}$ and feeding them to $f^b$ along with current feature maps $v_r$ makes $f^b$ learn accumulated knowledge. Please note that generator $\mathcal{G}$, an independent network, aids $f = \\{f^a, f^b, f_{[L]}\\}$, only to provide previous information. 2) encouraging benign forgetting means forgetting biased logic the model learned at previous tasks. Since self-supervised learning does not refer to label space, $f^b$ can represent bias-free feature space because bias is a spurious correlation between the feature and label space. Hence, it contributes to encouraging benign forgetting.
> We reflected this in Sec.4.1 and Sec.4.2 of the revision.
>
> > In the final step of Algorithm 1, and $f^{a}$ and $f^{b}$ will be retrained?
>
> Yes, they are retrained by cross-entropy loss to train $f^a$ and fit the feature space to the label space. One might argue that the final step of Algorithm 1 could overwrite the previous information learned with $v_f$ generated by $\mathcal{G}$. To prevent this issue, we 1) train $f^b$ by contrastive learning and 2) train $f = \\{f^a, f^b, f_{[L]}\\}$ by direct supervised learning following the iterative manner.
>
> > In practice, how to determine $l$ and $L$ in Section 4.2?
>
> We are sorry for not mentioning about $l$ setting, which is actually in Table 6 in the Appendix. We experimentally find the best $l$ for each dataset. Although the model performs the best when $l=2$ for MNIST, others still achieved competitive performance.

---

### Official Review · Reviewer_kjJH · 2022-10-26

**Confidence:** 3
**Correctness:** 4
**Technical Novelty And Significance:** 4
**Empirical Novelty And Significance:** 4
**Recommendation:** 8

**Clarity, Quality, Novelty And Reproducibility:**

- The problem is very novel.
- The proposed techniques look very effective.
- The paper is generally well written.


**Strength And Weaknesses:**

Strengths
- Defines and solves a novel and important problem.
- The problem is well formulated.
- The two key techniques are effective in handling malignant and benign forgetting and go well together.
- Experiments show clear benefits of the proposed method.

Weaknesses
- In Section 3.2, the explanation right after Equation (1) is a bit cryptic. Perhaps the authors can explain while making direct references to Figures 1 and 2? It also took a while to understand how $\beta$ plays a role when using the images in Figure 1.
- How generally applicable is Equation (1)? I don't think it's always the case that accuracy decreases as bias decreases. Usually one would expect lower bias to lead to higher accuracy.
- In Section 4.1, it is not clear why a GAN training on lower layers of a CNN discourages malignant forgetting only. Expanding the feature space seems to discourage both malignant and benign forgetting. What is the evidence that only malignant forgetting is affected? Also how does one set $l$?
- Why should contrastive learning be performed on $f_\theta^b$ instead of $f_\theta^a$ and $f_\theta^b$ combined (i.e., $f_{[1,\ldots,L-1]}$)?
- The contrastive learning seems effective when bias is defined as a correlation between label and attribute space. Although out of the scope of this paper, if bias is defined as some biased distribution in the attribute space, how would this technique perform?
- Section 4.2: "Since augmentations of input image space are not directly applicable in the feature space due to distribution shift" -> can you elaborate a bit more on why this is the case?
- In Section 5.2, please briefly explain how the baselines (LfF, EWC, LwF, DGR, GFR, ABD, HAL, DER) work.

**Summary Of The Paper:**

This paper coins the new problem of Learning without Prejudices in a continual learning (CL) setup. Unlike most CL works that assume catastrophic forgetting, which is considered bad, the paper is the first to point out that there are actually both malignant and benign forgettings and that one must actually encourage the benign forgetting. Benign forgetting happens when existing biases (correlations between the label and attributes) are gradually unlearned. Two key techniques are proposed: a GAN-like feature generation is used to discourage malignant forgetting by reducing dependencies to a few biased attributes and feature-level augmentation and contrastive learning, which encourages benign forgetting by bypassing the label-attribute correlation. Experiments show that the proposed method outperforms various baselines on several real datasets.

**Summary Of The Review:**

This paper proposes the novel problem of learning without prejudices by making the key observation that forgetting can be benign. The proposed techniques for encouraging benign forgetting and discouraging malignant forgetting are effective and go well with each other. Experimental results look strong. The presentation can be improved a bit.

---

> ### Author Response · Authors · 2022-11-17
> **Thanks for valuable and interesting comments (2/2)**
>
> > Contrastive learning seems effective when bias is defined as a correlation between label and attribute space. Although out of the scope of this paper, if bias is defined as some biased distribution in the attribute space, how would this technique perform?
>
> Thanks for the very interesting topic. Firstly, let an attribute $attr$ and another attribute $attr'$ have a significant but spurious correlation (hence, biased in attribute space). Then, there would be two cases for this issue: (a) they are also correlated to label $y$. (b) they are not correlated to $y$. As an example, for hair attribute in CelebA, length (short, long) would be very correlated to color (black, blond) as depicted in Figure 6, though hair length and color are not biologically correlated (so, spuriously correlated). One might argue that SSL (encouraging benign forgetting) would focus more on representing the correlation between attributes (maybe $attr$ and $attr'$ are entangled with each other in this case) due to the absence of the label information, potentially misleading the generalizability of the model.
>
> - If $attr$ and $attr'$ are natural meanings for labels, our model would use both of them. When $attr$ is the natural meaning corresponding to the target label but $attr'$ is a spurious correlation, it might be a problem. Because the model would use not only natural meaning $attr$ but also it's entangled $attr'$ out of necessity, potentially misleading generalization. We sincerely think that it could be a very interesting feature work.
> - It is desirable for the model not to employ both $attr$ and $attr'$ for prediction. However, although hair length and color are entangled in feature space, for instance, the model would not use them during optimization for cross-entropy loss because they are not helpful for prediction. Please note that the generalizability of LwP on CelebA quantitatively demonstrates our method is also robust in this scenario.
>
> > Section 4.2: "Since augmentations of input image space are not directly applicable in the feature space due to distribution shift" $\rightarrow$ can you elaborate a bit more on why this is the case?
>
> Firstly, augmentations for color (RGB) are limited to be applied to feature maps (e.g., grey scaling for 1024 channels does not work as we intended.). Second, for spatial augmentations, we often consider the visual characteristic of the images not to remove necessary information (e.g., we should not crop or cut out images randomly to train (with mask, without mask) classifier because it could erase mask in face images). However, We do not know what the feature maps represent exactly and hence cannot consider which one is appropriate.
>
> > In Section 5.2, please briefly explain how the baselines (LfF, EWC, LwF, DGR, GFR, ABD, HAL, DER) work.
>
> We are sorry for missing the presentation for baseline models in **Competing models**. Although Section **RELATED WORK** includes them, it is hard to match the competing models to them. For clarity, we evince all the baselines with their model name in **RELATED WORK** in revision.

---

> ### Author Response · Authors · 2022-11-17
> **Thanks for valuable and interesing comments (1/2)**
>
> > In Section 3.2, the explanation right after Equation (1) is a bit cryptic. Perhaps the authors can explain while making direct references to Figures 1 and 2?
>
> Thanks for this advice. We uploaded a modified revision with more clear explanation.
>
> > It also took a while to understand how plays a role when using the images in Figure 1.
>
> We are sorry for displaying a figure that is not intuitive to understand $\beta$. Please consider that to present (10 classes, 10 tasks, 10 colors, distorted test set) with $\beta$ might make the figure confusing. Nonetheless, with the same motivation as the reviewer, we specified the overall distribution of datasets in the Appendix as mentioned in the caption of Figure 1 "A more detailed dataset configuration is provided in the Appendix". Please consult Figures 4 - 7 in the appendix.
>
> > How generally applicable is Equation (1)? I don't think it's always the case that accuracy decreases as bias decreases. Usually one would expect the lower bias to lead to higher accuracy.
>
> This is valuable advice. This is the reason why we set the scaling factor, $Acc(f_\{t},\tilde{\mathcal{D}}_\{t})-\frac{1}{n(\mathcal{Y}_t)}$, as the denominator. If the model is already less-biased, the numerator $Acc(f_t, \tilde{\mathcal{D}}_t)$ $- Acc(f\_{t+1},\tilde{\mathcal{D}}_t)$ could become small, meaning the model does not need to forget that biased logic that much. Hence, by dividing the scaling factor, we can adjust BFR to increase when the model is biased severely at $t$ task and decrease when the model is less-biased.
>
> For application, BFR could include some other correlations (e.g., malignant forgetting). Hence, we do not think it could be used as an evaluation metric to estimate generalization. Though, we carefully suggest that it shows `forgetting phenomena could contribute to the generalization' providing the guideline for model design.
>
> > In Section 4.1, it is not clear why a GAN training on lower layers of a CNN discourages malignant forgetting only. Expanding the feature space seems to discourage both malignant and benign forgetting. What is the evidence that only malignant forgetting is affected?
>
> This feedback is very appropriate. The two goals of this study, *discouraging malignant forgetting* and *encouraging benign forgetting*, signify memorizing *previous information* and forgetting `biased logic', respectively. The biased logic is fundamentally learned by the model due to spurious correlations in a dataset. Therefore, if past information is memorized, different spurious correlations are unified to become less biased for this bias attribute (i.e., biased logic is mitigated). In this respect, GAN encourages benign forgetting, while discouraging malignant forgetting. We clarified this in the revision.
>
> Therefore, *discouraging malignant forgetting* is not perfectly independent of *encouraging benign forgetting*. However, please consider that generator $\mathcal{G}$ makes feature maps similar to past samples, contributing to memorizing previous information, and self-supervised learning helps the model to represent unbiased feature space, not using labels.
>
> > How does one set $l$?
>
> We are sorry for not mentioning about $l$ setting, which is actually in Table 6 in the Appendix. We experimentally find the best $l$ for each dataset. Although the model performs the best when $l=2$, others still achieved competitive performance.
>
>
> > Why should contrastive learning be performed on $f^b_{\theta}$ instead of $f^a_{\theta}$ and $f^b_{\theta}$ combined (i.e., $f_{[1, ..., L-1]}$) ?
>
> We generate feature maps $v_f$ including previous information by $\mathcal{G}$ and hence they cannot be fed to $f^a$ due to size. Therefore, intending to apply contrastive learning equally for $v_r$ and $v_f$, we train only $f_{\theta}^b$. However, the Image in Table 7 of the Appendix updates all the layers in the classifier $f_{\theta}$ with contrastive learning, meaning $l=0$.

---

### Official Review · Reviewer_K36S · 2022-10-26

**Confidence:** 5
**Correctness:** 3
**Technical Novelty And Significance:** 2
**Empirical Novelty And Significance:** 2
**Recommendation:** 5

**Clarity, Quality, Novelty And Reproducibility:**

The paper is generally clear; however, the scope of the problem is very narrow as the task share a common class set with the distribution of the test data being identical. The paper does not compare with more recent CUL papers and should compare it. It is not clear whether the experiment results are reproducible. Please refer to strengths and weaknesses.

**Details Of Ethics Concerns:**

None.

**Strength And Weaknesses:**

(+) The paper proposes an interesting problem. To categorize forgetting as malignant and benign and propose an algorithm to discourage malignant forgetting and encourage benign forgetting sound interesting. Essentially there is a distribution mismatch between the training and test dataset.
(-) In CL, the problem is generally defined assuming non-intersecting class labels; however, CUL is studied assuming that the class labels are equal. This restricted problem considered in this paper makes the CUL problem less interesting and very restrictive.  Since the test data in this paper for every task have same distribution, there does not seem to be any catastrophic forgetting in the usual sense.
(-) The readability of the paper could be improved with examples and figures.
(-) Comparative experimental results do not include more recent CL algorithms and the datasets used for evaluation are small and not extensive.


**Summary Of The Paper:**

The paper introduces concepts of benign forgetting and malignant forgetting in continual learning, and it conducts a study to discourage malignant forgetting and encourage benign forgetting. The performance of the proposed method is compared to previously proposed continual learning algorithms.

**Summary Of The Review:**

The paper is generally clear; however, the scope of the problem is very narrow as the task share a common class set with the distribution of the test data being identical. The paper does not compare with more recent CUL papers and should compare it. It is not clear whether the experiment results are reproducible.

---

> ### Author Response · Authors · 2022-11-17
> **Thanks for fundamental and helpful advice**
>
> > Restricted problem setup due to the constant number of classes.
>
> We admit that equal class labels are a limited condition compared to other conventional scenarios, e.g., class-incremental learning. However, we carefully suggest that it is still valuable in practice, where the model is deployed for one specific task for a long period of time, updated with newly collected data. For instance, please consider a defection-detecting model classifying (good(0), bad(1)). It would be better for the machine to be updated periodically with new data, but they could have a different bias due to various conditions affecting them (e.g., humidity, temperature, dust concentration in the factory, and so on).
>
> Nonetheless, we really consent with the reviewer and it would be interesting and necessary if conventional CL scenarios are combined with CUL (e.g., class-incremental unbiased learning).
>
> > Since the test data in this paper for every task have same distribution, there does not seem to be any catastrophic forgetting in the usual sense.
>
> Unbiased average accuracy $\overline{Acc}_{ub}$ in Sec. 2 is calculated with one test data. In this evaluation protocol, as the reviewer pointed out, malignant forgetting might not be meaningful. However, since multiple validation sets following the same distribution as each task are used to estimate average accuracy $\overline{Acc}$ in Sec.2, addressing malignant forgetting is still important. Please consult with this metric.
>
> Nonetheless, please note that what we actually intended in this paper is to make the model focus on natural meaning corresponding to the target label for prediction (e.g., the shape of the number for Biased MNIST). Our motivation is that the framework referring to natural meaning is a more desirable and robust approach for predicting both validation and test sets correctly.
>
> > The readability of the paper could be improved with examples and figures.
>
> Thanks for this advice. For better presentation, we added the figure of Biased CIFAR-10 and Biased CelebA-HQ. Please consult Figure 1 in Appendix.
>
>
> > Comparative experimental results do not include more recent CL algorithms and the datasets used for evaluation are small and not extensive.
>
> Consenting with the reviewer's comment, we added LiDER (Neurips2022) [1] to our baseline models. We use the LiDER model based on ER-ACE [2]. Please consult Table 1 in the script.
>
> > It is not clear whether the experiment results are reproducible.
>
> Please consult with the **B. EXPERIMENT DETAIL** in the Appendix for the reproduction. And, please consult our source code in the Supplementary Material.
>
> > References
>
>
> [1] Bonicelli, *et al*. On the Effectiveness of Lipschitz-Driven Rehearsal in Continual Learning. Advances in Neural Information Processing Systems 35. 2022
>
> [2] Lucas Caccia, *et al*. New Insights on Reducing Abrupt Representation Change in Online Continual Learning. In International
> Conference on Learning Representations Workshop, 2022.

---

### Official Review · Reviewer_vbj9 · 2022-11-01

**Confidence:** 4
**Correctness:** 3
**Technical Novelty And Significance:** 2
**Empirical Novelty And Significance:** 2
**Recommendation:** 6

**Clarity, Quality, Novelty And Reproducibility:**

The paper is clearly presented.
The quality can be further improved especially on the experiment part.
The novelty seems not very strong due to the reasons mentioned in ``Strength And Weaknesses".


**Strength And Weaknesses:**

Strength:
The setting and method are clearly presented.
The effectiveness of the proposed method under the proposed setting is demonstrated in several synthetic datasets.

Weaknesses:
The setting of the paper seems artificial. Specifically, this work considers the case where strong correlations (large beta) exists between the label space and feature space. And each task contains data with different yet strong correlations. And the training is under continual learning case where old data cannot be re-visited hence strong correlations cannot be regularized using other data. Though I agree that this setting is new, but it is not very convincing to me that it is practical. For example, if we know that each task has strong bias, why would we completely drop previous data?

In the meantime, the algorithm design seems particularly-biased to the proposed setting, and not very unexpected. Specifically, in the proposed setup, each task has a strong correlation between labels and features, hence using self-supervised learning where the model is regularized without labels seems somewhat natural to me, making it less surprising. Though for this point, I agree that at least to combine GAN and BYOL, there are some technical contributions made in this work.

Moreover, all experiments are conducted with synthetic data, in small scale (MNIST/CIFAR level, with < 10 tasks), and with strong artificial bias. I agree on the motivated examples that maybe frogs are taken frequently with swamp, but I can hardly image that in practice similar cases happens as in the proposed setting, for example, frog sometimes mostly taken with swamp, yet later taken with sky etc. Hence, I believe to better motivate the practicality of the proposed setting and the effectiveness of the proposed methods, two things need to be done:
1) Show that in practice, the proposed setup actually happens, using real-world datasets.
2) In practice, it is at least often unknown whether strong biases occurs in all tasks in the sequence. Hence, the proposed method must be also effective when no strong bias is presented in continual learning, a fair comparison against other CL methods is required on such setting (maybe normal CL problems).

Also, there is no notion of computation/parameter increase during the experiments. It would be clearer if the increase on the model parameters and computations during training can be explicitly reported, since two new branches are required for GAN and BYOL. And maybe for fairer comparisons, one can compare with baseline models with the same model size etc.. But I do understand that it might be hard to compare completely fairly on this point.

Finally, for replay-based methods, only 200 images are used in the buffer, I wonder whether all methods are still much worse than the proposed one when the replay buffer is enlarged? Say at least 10 times larger?

**Summary Of The Paper:**

This paper proposes the setting of ``continual unbiased learning" (CUL), where for each task of continual learning, labels are strongly correlated with specific but different input features. It also proposes to use self-supervised learning methods to address the model bias issue during CUL.

**Summary Of The Review:**

As explained in detail in ``Strength And Weaknesses". I agree that the paper proposes a new setting and effective method under this setting. However, according to the weaknesses part, I think the paper needs further motivation of the proposed setting and supporting evidence of the method effectiveness in general. And the novelty is somewhat limited.

---

> ### Author Response · Authors · 2022-11-17
> **Thanks for many suggestions to improve the paper (2/2)**
>
> > For replay-based methods, only 200 images are used in the buffer.
>
> Thanks for this advice. We agree that learning with a replay buffer helps generalization in CUL. Yet, there are some special cases, where access to prior samples is strictly limited (e.g., privacy data such as medical or financial data, with a short storage period). In this point of view, we carefully suggest that the learning scenario without prior samples can be deployed in more diverse situations. However, please consider that the replay buffer can be employed with our method, showing competitive generalization as depicted in the following Table `Continual Unbiased Learning with Replay Buffer'. Further, we additionally investigated the generalizability of the model with a larger buffer (2,000) following the reviewer's advice and report it. Please note that our method also achieves the best generalizability compared to replay-based models for small (200) and large (2,000) buffers.
>
> |                | MNIST |  |      CIFAR-10            |                  |
> |:--------------:|:-------------------------:|:----------------------------:|:----------------:|:----------------:|
> |                | 200                       | 2,000                        | 200              | 2,000            |
> | HAL   | 82.40($\pm$2.21)          | 84.28($\pm$0.75)             | 23.50($\pm$8.00) | 29.24($\pm$0.48) |
> | DER   | 90.37($\pm$1.33)          | 89.84($\pm$0.72)             | 31.07($\pm$0.53) | 34.15($\pm$0.21) |
> | LiDER | 89.44($\pm$6.07)          | 92.61($\pm$2.14)             | 28.02($\pm$0.48) | 35.72($\pm$0.76) |
> | LwP   | 92.34($\pm$0.55)          | 93.14($\pm$0.95)             | 32.06($\pm$0.59) | 40.18($\pm$0.37) |

---

> > ### Comment · Reviewer_vbj9 · 2022-11-22
> > **Response**
> >
> > Thanks for your efforts to conduct extra experiments. I think most of my concerns are addressed properly. The remaining questions are 1) that it would be better to find non-synthetic datasets to validate the effectiveness of the proposed algorithm. 2) Does the proposed method introduce heavy extra computation on training self-supervised/GAN branches.
> >
> > But I would slightly raise my score since other questions are mostly addressed.

---

> > > ### Author Response · Authors · 2022-12-07
> > > **Many thanks**
> > >
> > > Many thanks for your response and re-evaluation. We are glad to hear that we addressed most of your concerns.
> > >
> > > We carefully answer the rest of the questions as follows:
> > >
> > > 1) We agree that further evaluation on the non-synthetic dataset is better to validate the generalizability of the proposed method (despite the Biased CelebA). It would be valuable to make real-world datasets for CUL in future work. We sincerely thank you for this suggestion.
> > >
> > > 2) We found that our proposed method uses about 20% more memory cost than the baseline method, which is enough competitive to other SOTA models as depicted in the following table. The values in the table mean the ratio of the amount of memory used to the baseline model.
> > >
> > > |        | Base | EWC | LwF | DGR | GFR | ABD | LfF | HAL | DER | LiDER | LwP |
> > > |:------:|:----:|:---:|:---:|:---:|:---:|:---:|:---:|:---:|:---:|:-----:|:---:|
> > > | Memory |  $\times$1.0 | $\times$9.4 | $\times$1.3 | $\times$1.3 | $\times$1.1 | $\times$1.4 | $\times$1.2 | $\times$1.2 | $\times$1.1 |$\times$1.7 |$\times$1.2 |
> > >
> > > For the computation, we admit that (generator + SSL) model could increment cost. Nonetheless, our model exhibits a similar cost to conventional generator-based CL models because the online network for SSL is part of the backbone network (classifier) and only momentum update is applied. As momentum update does not need gradient-descent operations, subtle cost is added.
> > >
> > > Thanks again for your efforts in the review process.

---

> ### Author Response · Authors · 2022-11-17
> **Thanks for many suggestions to improve the paper (1/2)**
>
> > The proposed method must be also effective when no strong bias is presented in continual learning.
>
> We admit that the experimental datasets (MNIST, CIFAR) might be limited to covering many cases of the real world. We are firstly really sorry for this issue. However, please consider that constructing biased sets, especially differently biased real-world sets for all the tasks, is not easy. This is because very few datasets have labels for various attributes. But, we could sample the data using 40 attributes provided by CelebA, a real-world face image dataset, and found significant correlations (gender, hair color), (gender, hair length), (gender, makeup). Please consult with the CelebA in Table 1 to compare LwP with baseline models for real-world datasets.
>
> > The proposed method must be also effective when no strong bias is presented in continual learning.
>
> We consent to the reviewer for the results reported in Table 1 ($\beta = 0.95$). To address this issue, we compared LwP for the less-biased set ($\beta = 0.8, 0.85, 0.9, 0.95$) to all the competing models as depicted in Figure 3. Nonetheless, for better justification, we widened the range of $\beta$ from $0.8 \sim 0.95$ to $0.1 \sim 0.95$. It is notable that our method generalizes best for all the $\beta$. Please consult Figure 3 in the main script and Figure 9 in the Appendix, respectively. We add this experimental result in the Appendix of the revision.
>
> > There is no notion of computation/parameter increase during the experiments.
>
> We really consent with the reviewer because the more parameters the model has, the better performance the model exhibits, generally. And, thanks for the consideration, "I do understand that it might be hard to compare completely fairly on this point.". As the reviewer pointed out, a simple comparison for the number of parameters is not fair because many of the methods for CL have addressed catastrophic forgetting by regularization-based (additional operation for regularization) and replay-based (more training iterations with buffer) approaches, which are not related to model size. Nonetheless, generator-based methods can be compared for model size. Therefore, we reduced the parameters of LwP and evaluated the generalizability. The following two Tables show our model still has competitive performance. We sincerely thank the reviewer for pointing out this issue because we could have an opportunity to understand our model and develop a new light version of LwP that have still competitive performance.
>
> |               | Dataset                            | DGR     | GFR     | ABD     | LwP     | LwP(tiny) |
> |:-------------:|:----------------------------------:|:----------------:|:----------------:|:----------------:|:----------------:|:------------------:|
> | $\mathcal{G}$ | MNIST    | 2.30 M           | 0.44 M           | 6.50 M           | 6.52 M           | 0.38 M             |
> | $\mathcal{D}$ |                                    | 2.76 M           | 0.34 M           | -                | 2.77 M           | 0.18 M             |
> | Total         |                                    | 5.06 M           | 0.78 M           | 6.50 M           | 9.29 M           | 0.56 M             |
> | $\overline{Acc}_{ub}$      |                                    | 85.04($\pm$1.32) | 77.50($\pm$2.78) | 82.78($\pm$1.05) | **91.57($\pm$0.82)** | *91.33($\pm$0.63)*   |
> | G             | CIFAR-10 | 3.58 M           | 0.63 M           | 8.42 M           | 8.46 M           | 2.07 M             |
> | D             |                                    | 2.76 M           | 0.54 M           | -                | 2.82 M           | 0.72 M             |
> | Total         |                                    | 6.34 M           | 1.17 M           | 8.42 M           | 11.28 M          | 2.79 M             |
> | $\overline{Acc}_{ub}$      |                                    | 28.94($\pm$2.61) | 29.11($\pm$0.48) | 29.67($\pm$0.61) | **31.18($\pm$0.29)** | *30.65($\pm$1.73)*   |

---

### Author Response · Authors · 2022-11-17
**Many thanks to all the reviewers for thoughtful comments**

We sincerely thank the reviewers for providing valuable comments and suggestions to improve the script. Consenting to the reviewers' advice, we tried to revise our paper with different 6 colors, which are reviewer1 (red), reviewer2 (orange), reviewer3 (green), reviewer4 (blue), multi-reviewers (purple), and ourselves (brown). The list we modified is as follows:

- Experimental results with Biased CIFAR10 for some ablation studies (Section C.1, C.2 in Appendix)
- More understandable presentation for benign forgetting rate (Section 3.2 in the script)
- Clarification of the model design for addressing forgetting (Section 4.1, 4.2 in the script)
- Addition of a more recent model for comparison (Table 1 in the script)
- Clarification for $l$ setting (Section 5.2 in the script)
- Clarification for baseline models (Section 6 in the script)
- Additional experiment for model size (Section C.3 in Appendix)
- Additional experiment for replay buffer (Section C.4 in Appendix)
- Investigation on more various $\beta$ (Section C.5 in Appendix)
- Description of BYOL (Section D in Appendix)

We tried to understand and respond to the reviewers' comments as well as possible. Please consider the following responses.

---

> ### Author Response · Authors · 2022-11-18
> **Modify the figures**
>
> We additionally modified Figure 3 in the main script and Figure 9 in the Appendix, respectively. We added the LiDER (NIPS 2022) model to the figures.

---

### Decision · Program_Chairs · 2023-01-20

**Decision:**

Accept: poster

**Justification For Why Not Higher Score:**

The paper has enough merits to be published as it is proposing an important new problem definition which is technically sound and applicable. Moreover, it also proposes a sensible approach to address it with a complete experiment. However, the impact is still somewhat limited as the experiments are more of a sanity check validating the approach without any realistic data. Finding realistic data for CL is not an easy task and I believe it does not invalidate the paper. However, it clearly invalidates score greater than poster.

**Justification For Why Not Lower Score:**

The proposed problem is important for the community to learn and discuss. Moreover, there is no major issue in algorithm development or empirical study. Hence, I believe it is ready to be shared with the community.

**Metareview: Summary, Strengths And Weaknesses:**

The paper is proposing a new and interesting problem continual learning with biased data. This is clearly an interesting extension of existing CL approaches and OOD/DomainGeneralization approaches. However, it is much more than a simple combination, as the whole is much more than some of parts here. Bias typically has some strong temporal/dynamic component and using it within the framework of CL is sensible. Moreover, authors propose a technically sound approach using feature generation and contrastive learning. The paper is reviewed by four experts and three of them agrees on the merits. Moreover, all reviewers unanimously agree on the novelty.

**Note From Pc:**

if the above contains the word "oral" or "spotlight" please see: "oral" presentation means -> notable-top-5% and "spotlight" means -> notable-top-25%. As stated in our emails, we are disassociating presentation type from AC recommendations